# Addressing Challenges in Reinforcement Learning for Recommender Systems with Conservative Objectives

## Abstract

Attention-based sequential recommendation methods have shown promise in accurately capturing users' evolving interests from their past interactions. Recent research has also explored the integration of reinforcement learning (RL) into these models, in addition to generating superior user representations. By framing sequential recommendation as an RL problem with reward signals, we can develop recommender systems that incorporate direct user feedback in the form of rewards, enhancing personalization for users.

Nonetheless, employing RL algorithms presents challenges, including off-policy training, expansive combinatorial action spaces, and the scarcity of datasets with sufficient reward signals. Contemporary approaches have attempted to combine RL and sequential modeling, incorporating contrastive-based objectives and negative sampling strategies for training the RL component. In this work, we further emphasize the efficacy of contrastive-based objectives paired with augmentation to address datasets with extended horizons. Additionally, we recognize the potential instability issues that may arise during the application of negative sampling. These challenges primarily stem from the data imbalance prevalent in real-world datasets, which is a common issue in offline RL contexts. Furthermore, we introduce an enhanced methodology aimed at providing a more effective solution to these challenges. Experimental results across several real datasets show our method with increased robustness and state-of-the-art performance. Our code is available via sasrec-ccql

## 1 Introduction

Recommender systems (RS) have become an indispensable tool for providing personalized content and product recommendations to users across various domains, such as e-commerce (Chen et al., 2019b), social media (Jiang et al., 2016), and news (Zhu et al., 2019). User-item interactions usually unfold sequentially, both the timing and order of these interactions are critically important. Early approaches to sequential recommendation are mainly powered by recurrent neural network (RNN)-based models (Wu et al., 2017; Yu et al., 2016). Later, models like Transformers (Vaswani et al., 2017) have enhanced understanding of user preferences in behavior sequences, improving recommendation accuracy (Zhou et al., 2020). It is worth noting that large language models such as GPT-4 (OpenAI, 2023), which have architectural similarities to RS transformers, have been shown to perform well at item recommendation tasks in a zero-shot fashion (Li et al., 2023).

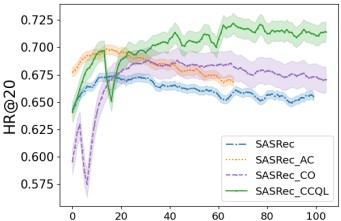

Figure 1: Enhanced stability and performance on RetailRocket purchase prediction with SASRec-CCQL, an approach that combines contrastive learning and RL-based objectives.

Despite this progress, sequence modelling is only part of the problem: it is also crucial to optimize the recommendation strategy itself. Reinforcement learning (RL) offers an appealing framework for this purpose, as it enables the recommender system to learn an optimal policy through interaction with the environment, balancing the trade-off between exploration and exploitation (Christakopoulou et al., 2022b; Chen, 2021). By incorporating RL into

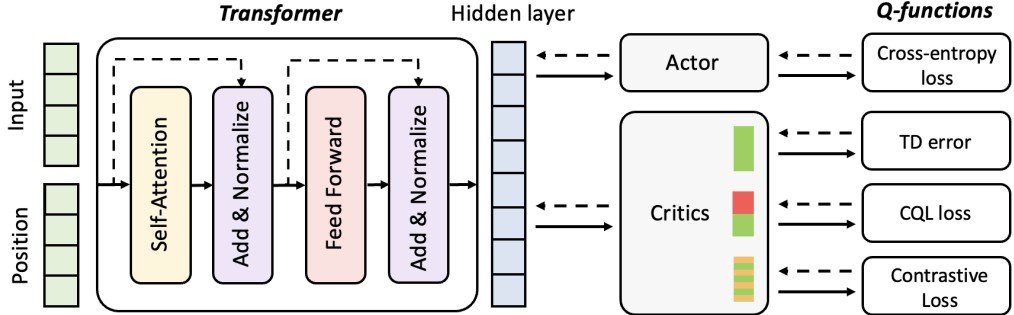

Figure 2: Model architecture for the training process and the interaction between the transformer model and Q-learning with the proposed objectives. The Conservative Q-learning (CQL) objective considers positive samples (green) and hard negative action sampling (red), while the contrastive objective is applied batch-wise across different user items (green vs orange). For more details refer to Sec.3.

the recommendation process, the system can actively adapt to changing user preferences and item catalogs, maximizing long-term user satisfaction rather than merely focusing on immediate rewards.

While RL provides an ideal framework to capture user preferences, it does not inherently solve one of the challenges in recommendation systems: high instability during training (Tang et al., 2023), also as evident in Figure 1, a particularly acute problem with larger, more complex models.

In this work, we propose to address this instability using two different components: contrastive learning and conservative Q-learning, as outlined in Figure 2. The components jointly encourage robust representation learning, improving performance and stability. Our extensive experimentation across multiple datasets demonstrates our method not only enhances the precision of recommendations in comparison to the baseline, but also adds further stability to the training process.

Our core contributions are as follows:

- We investigate the application of a contrastive learning objective, in conjunction with sequential augmentation strategies, and provide empirical evidence of its effectiveness on a variety of challenging real-world datasets.
- We pinpoint fundamental problems arising from the inclusion of negative action sampling, as proposed in previous studies, and propose a more conservative objective to alleviate these instabilities during RL training.
- Our analysis underscores the need to monitor training progress for RL-based models to detect instabilities that could impair model performance in online deployment. We advocate for the reporting of training progression in parallel with tabular results in the application of reinforcement learning.

## 2 RELATED WORK

**Sequential recommendation** Sequential recommendation aims to capture users interests based on their historical behaviors. Earlier work focused on latent representation methods (Choi et al., 2012; Zhao et al., 2013) and Markov chain models (Rendle et al., 2010). With the introduction of deep learning, Convolutional Neural Networks (Tang & Wang, 2018; Yuan et al., 2019), Recurrent neural networks (Wu et al., 2017; Yu et al., 2016) and graph neural networks (Chang et al., 2021; Ying et al., 2018) have become popular and powerful backbone models for recommender systems. The success of Transformer models in sequence modeling tasks across different fields has led to their combination with RL in sequential recommendation tasks (Xin et al., 2020; 2022; Zhao et al., 2018; Sun et al., 2019; Zhou et al., 2020). SASRec (Kang & McAuley, 2018) adapts transformers to next-item prediction in recommender systems. The transformer architecture utilized in this work leverages its self-attention function to assign weights to different items in the user's history, effectively

identifying the items most relevant to the user's current situation. Sun et al. (2019) employed BERT to enhance recommendation precision and personalization. BERT4Rec (Zhou et al., 2020) incorporates bidirectional encoder representations from transformers, considering that sequential recommendations may not strictly adhere to the ordering assumptions in language models.

**RL for sequential recommendation** RL allows recommender systems to model sequential, dynamic user-system interactions while considering long-term user engagement (Afsar et al., 2022). In this framework, the recommender system learns to interact with its environment (users and items) by executing actions (offering recommendations) and observing the subsequent rewards (user feedback) to refine its strategy over time. Christakopoulou et al. (2022a) developed their methodology based on the REINFORCE (Sutton & Barto, 2018) algorithm, emphasizing the role of reward shaping in aligning the objectives of the RL recommender with user preferences. They evaluate their method using proprietary data and incorporate a satisfaction imputation network for assessing user-item interactions. Chen et al. (2019c); Bai et al. (2019) attempt to eliminate the off-policy issue by building a model to imitate user behavior dynamics and learn the reward function. The policy can then be trained through interactions with the simulator. ResAct (Xue et al., 2023) proposes Residual Actor which starts by imitating the online serving policy and subsequently adding an action residual to arrive at a policy. However, our method adopts a fundamentally different strategy which instead of learning residuals, our policy is designed to be fully predictive, directly outputting discrete actions given a state.

**Contrastive learning for recommendation** Contrastive learning aims to learn a data representation by bringing similar instances closer together in the representation space while pushing dissimilar instances farther apart. Although contrastive learning has been widely studied and demonstrated remarkable performance in computer vision (He et al., 2020; Chen et al., 2020) and natural language processing (Gao et al., 2021; Liu et al., 2021a), it is under-explored in recommendation systems. CL4SRec (Xie et al., 2020) integrates contrastive learning objectives within the SASRec framework. While their assessment is carried out on recommendation datasets, they do not take into account datasets based on rewards, nor do they incorporate reinforcement learning in their approach. A graph contrastive learning model (Liu et al., 2021b) learns the embeddings in a self-supervised manner and reduces the randomness of message dropout. This graph contrastive model has been integrated with several matrix factorization and GNN-based recommendation models.

**Training stability in recommendation systems** Training instability (Gilmer et al., 2021) presents a significant challenge, particularly when the loss diverges instead of converging. This in turn yields models that are more prone to have training instability when the model is large or complex. Limited research has been conducted to address training stability in recommendation models. However, a recent study by Tang et al. (2023) tackles this issue by improving the loss optimization landscape, enhancing stability in real-world multitask ranking models, such as YouTube recommendations.

## 3 METHOD

Let $I$ denote the item set, then a user-item interaction sequence can be represented as $x_{1:t} = \{x_1, x_2, ..., x_{t-1}, x_t\}$, where $x_i \in I$ ($0 < i \leq t$) denotes the interacted item at timestamp $i$ at time step $t$. The task of next-item recommendation is to recommend the most relevant item $x_{t+1}$ to the user, given the sequence of $x_{1:t}$. A common solution is to build a recommendation model whose output is the classification logits $y_{t+1} = [y_1, y_2, ..., y_n] \in \mathbb{R}^n$, where $n$ is the number of candidate items. Each candidate item corresponds to a class. The recommendation list for timestamp $t + 1$ can be generated by choosing top-$k$ items according to $y_{t+1}$. Typically one can use a generative sequential model $G(\cdot)$ to encode the input sequence into a hidden state $s_t$ as $s_t = G(x_{1:t})$. Given an input user-item interaction sequence $x_{1:t}$ and an existing recommendation model $G(\cdot)$, the supervised training loss is defined as the cross-entropy over the classification distribution: $\mathcal{L} = -\frac{1}{|N|} \sum_{i \in N} \sum_{c \in C} y_{i,c} \log(p_{i,c})$ where, $|N|$ is the cardinality of the set $\mathcal{N}$, the term $y_{i,c}$ is 1 if the user interacted with the $i$-th item and 0 otherwise, and $p_{i,c}$ is the model's estimated probability.

### 3.1 RECOMMENDATION AS AN RL PROBLEM

Viewing the recommendation problem through the lens of RL offers a different perspective on modeling user preferences and optimizing recommendation strategies. In this framework, the

recommender system learns to interact with its environment (users and items) by executing actions (offering recommendations) and observing the subsequent rewards (user feedback) to refine its strategy over time. The system's objective is to determine which content or product to recommend to incoming user requests, considering factors such as user profiles, context, and interaction history. To achieve this, the recommendation problem is formulated as a *Markov Decision Process*, represented by the tuple $\langle \mathcal{S}, \mathcal{A}, P, R \rangle$ with state space $\mathcal{S}$ and action space $\mathcal{A}$. Actions $a \in \mathcal{A}$ correspond to the items available for recommendation, while states $s \in \mathcal{S}$ represent user interests in the form of items they interact with. $P$ denotes the latent transition distribution capturing $s_{t+1} \sim \mathbf{P}(.|s_t, a_t)$ i.e. how user state changes from $t$ to $t + 1$, conditioned on $a_t$ and $s_t$. Lastly, the reward $r(s, a)$ represents the immediate reward obtained by performing action $a$ for state $s$. The goal is to find a policy $\pi(a|s)$ that represents probability distribution over the action space, (i.e. items to recommend given the current user state $s \in \mathcal{S}$) which maximize the expected cumulative reward $\max_\pi \mathbf{E}_{\tau \sim \pi}[R(\tau)]$, where $R(\tau) = \sum_{t=0}^{|\tau|} r_t$, and the expectation $\mathbf{E}$ is taken over user trajectories $\tau$ obtained by acting according to the policy $a_t \sim \pi(.|s_t)$ and transition dynamics $s_{t+1} \sim \mathbf{P}(.|s_t, a_t)$.

Following the approach in Xin et al. (2020), the Transformer model, $G(\cdot)$, encodes the input sequence into a latent state $s_t$ which is then reused as the state mapping for the reinforcement learning model. This sharing schema of the base model enables the transfer of knowledge between supervised learning and RL. The loss for the reinforcement learning component is defined based on the one-step Temporal Difference (TD) error (Sutton, 1988) :

$$L_Q = \mathbb{E}[(r(s_t, a_t) + \gamma \max_{a_{t+1}} Q(s_{t+1}, a_{t+1}) - Q(s_t, a_t))^2] \tag{1}$$

The TD error is computed as the discrepancy between the estimated $Q$-value and the sum of the actual observed reward and the discounted estimated $Q$-value of the following state-action pair. The supervised loss and the reinforcement learning loss are jointly trained in this framework. This optimization step refines the critic's parameters, enhancing its capability to estimate the state-action value function Q(s, a). By integrating the supervised learning signal, the critic benefits from additional guidance during training, leading to more accurate Q-value estimations.

The above constitutes the core of our learning algorithm. In what follows, we expand on this framework by introducing two families of improvements: (1) we utilize conservative Q-Learning (Kumar et al., 2020) to mitigate issues related to off-policy training and (2) we introduce a contrastive learning objective to further improve the quality of learned representations.

## 3.2 Negative Action Sampling

As previously discussed, the implementation of RL algorithms within RS settings presents challenges in relation to off-policy training and an insufficient number of reward signals. In an offline RL setting, it's generally assumed that a static dataset of user interactions is available. The principal challenge in offline RL involves learning an effective policy from this fixed dataset without encountering the problems of divergence or overestimation. This issue is further compounded by the inadequate presence of negative signals in a typical recommendation dataset. Relying solely on positive reward signals such as clicks and views, while ignoring negative interaction signals, can result in a model that exhibits a positive bias. To address this, Xin et al. (2022) introduced a negative sampling strategy (SNQN) for training the RL component. The authors further propose Advantage Actor-Critic (SA2C) for estimating Q-values by utilizing the "advantage" of a positive action over other actions. Advantage values can be perceived as normalized Q-values that assist in alleviating the bias arising from overestimation of negative actions on Q-value estimations. This is then combined with a propensity score to implement off-policy correction for off-policy learning. Propensity scoring is a statistical technique often used in observational studies to estimate the effect of an intervention by accounting for the covariates that predict receiving the treatment. In the context of RL, the propensity score of an action is often equivalent to the probability of that action being chosen by the behavior policy (Chen et al., 2019a). The use of propensity scores for off-policy correction in reinforcement learning has similarities with importance sampling (IS). Both techniques aim to correct for the difference between the data-generating (behavior) policy and the target policy. IS is a technique used to estimate the expected value under one distribution, given samples from another. IS uses the ratio of the target policy probability to the behavior policy probability for a given action as a weighting factor in the update rule.

### 3.3 Conservative Q-Learning

There are potential issues associated when using propensity scores or IS for off-policy corrections. One concern is the high variance of IS, particularly when there is a significant disparity between the target policy and the behavior policy. This occurs because the IS ratio can become excessively large or small. The propensity score approach can encounter similar challenges. Consequently, the high variance can introduce instability in the learning process, ultimately resulting in divergence of the Q-function, as depicted in Figures 3 and 4.

We posit that estimating both the advantage function and propensity scores can introduce bias if they are not accurately computed. This bias can arise from function approximation errors, estimation errors, or modeling errors. Moreover, the aforementioned figures provide evidence that high variance in the estimated advantage function or propensity scores can lead to instability and potential divergence. Such instability may stem from over-optimistic Q-value estimates, representing a specific instance of learning process instability. Overestimated Q-values can lead to erroneous learning and subpar policy performance, as empirically demonstrated in Section A.1.

Conservative Q-Learning (CQL) (Kumar et al., 2020) is designed to address the overestimation issue commonly associated with Q-learning. In CQL, a conservative value function is employed to estimate the optimal action-value function. This conservative value function is defined as the minimum of the current estimate and the maximum observed return for a given state-action pair. The primary concept underlying CQL involves minimizing an upper bound on the expected value of a policy, taking into account both in-distribution actions (actions present in the dataset) and potential out-of-distribution actions. This is achieved by minimizing the following objective:

$$\mathcal{L}_{\text{CQL}}(\theta) = \mathbb{E}_{(s,a,r,s') \sim \mathcal{D}} \Big[ \big( Q_\theta(s,a) - r - \gamma \mathbb{E}_{a' \sim \pi_{\phi(a'|s')}} [Q_{\theta'}(s',a')] \big)^2 \Big] +$$
$$\alpha \, \mathbb{E}_{s \sim \mathcal{D}} \Big[ \log \sum_a \exp Q_\theta(s,a) - \mathbb{E}_{a \sim \hat{\pi}_\beta(a|s)} [Q(s,a)] \Big] \tag{2}$$

Here, $\mathcal{D}$ represents the fixed dataset, $\theta$ and $\theta'$ are the parameters of the Q-function and its target network, $\phi$ is the policy parameters, $\gamma$ is the discount factor, $\alpha$ is a temperature parameter that controls the trade-off between Q-function minimization and the conservative regularization. In our experimental setting, in contrast to the original formulation of CQL where the Q-function is assessed on random actions, we evaluate the Q-function explicitly on negative actions. These actions correspond to items with which the user has never interacted with. This approach is predicated on the assumption that such missing interactions represent a set of items in which the user has no interest. In scenarios where further user-item interaction is possible, uncertainties can be mitigated by gathering more representative data distinguishing liked and disliked items for each user.

### 3.4 Contrastive Learning with Temporal Augmentations

InfoNCE (Noise Contrastive Estimation) (van den Oord et al., 2018), a commonly used loss function in contrastive learning, helps in learning effective representations. The objective is computed using positive sample pairs $(x_j, y_j)$ and a set of negative samples $y_{j,k}$.

$$\mathcal{L}_{\text{InfoNCE}} = -\frac{1}{M} \sum_{j=1}^{M} \log \frac{\exp\left(f(x_j, y_j)\right)}{\exp\left(f(x_j, y_j)\right) + \sum_{k=1}^{K} \exp\left(f(x_j, y_{j,k})\right)} \tag{3}$$

where $M$ is the number of positive sample pairs, $K$ is the number of negative samples for each positive pair, and $f(x_j, y_j)$ is the similarity function (e.g., dot product in the embedding space) between the representations of $x_j$ and $y_j$, $f(x_j, y_{j,k})$ measures the similarity between $x_j$ and a negative sample $y_{j,k}$. The goal of the InfoNCE loss is to maximize the similarity between positive pairs while minimizing the similarity between negative pairs, thus learning useful representations in the process.

To boost model performance, we found combining it with contrastive learning to be most beneficial. Empirical analysis shows both methodologies can effectively use offline data - contrastive learning for representation learning and CQL for policy/value learning. This method is particularly useful in scenarios where online interaction is costly or impractical, as it is our case with a recommender system using a static dataset. The overall objective to optimize becomes:

$$\mathcal{L} = \mathcal{L}_{CE} + \omega\mathcal{L}_Q + \mathcal{L}_{CO} + \alpha\mathcal{L}_{CQL} \tag{4}$$

where $\mathcal{L}_{CE}$ is the cross-entropy loss, $\mathcal{L}_Q$ is the Q-learning i.e. TD loss, $\mathcal{L}_{CO}$ is the contrastive objective and $\mathcal{L}_{CQL}$ is the conservative Q-learning objective. Figure 2 depicts our proposed architectural framework, SASRec-CCQL.

## 4 EXPERIMENTS

We conducted experiments on five real-world datasets to evaluate the performance of our methods, namely **SASRec-CO** and **SASRec-CCQL**. Our experiments aim to address the following questions:

**Q1**: How does the framework perform when integrated with the proposed objectives?

**Q2**: How do different negative sampling strategies impact performance and, more importantly, the stability of RL training? How do solutions like SA2C versus SASRec-CCQL mitigate some of these instability issues?

**Q3**: To what extent does the performance improvement stem from the integration of RL, and what are the effects of short-horizon versus long-horizon reward estimations?

### 4.1 DATASETS, BASELINES AND EVALUATION PROTOCOLS

**Datasets.** We use the following five real world dataset: **RetailRocket** (Kaggle, 2017): Collected from a real-world e-commerce website, it contains sequential events corresponding to viewing and adding to cart. The dataset includes $1,176,680$ clicks and $57,269$ purchases over $70,852$ items. **RC15** (Ben-Shimon et al., 2015): Based on the dataset of the RecSys Challenge 2015, this session-based dataset consists of sequences of clicks and purchases. The rewards are defined in terms of buy and click interactions. **Yelp** (Yelp, 2021): This dataset contains users' reviews of various businesses. User interactions, such as clicks or no clicks, are interpreted as rewards. **MovieLens-1M** (Harper & Konstan, 2015): A large collection of movie ratings. **AmazonFood** (Majumder et al., 2019): This dataset consists of food reviews from Amazon which we only use the last two datasets with non-RL-based baselines since there are no rewards in these two datasets, and our focus is solely on observing the benefits of contrastive learning objectives.

**Baselines.** We compare our method to a range of baselines from the code released by (Xin et al., 2020; 2022). Please refer to (Xin et al., 2020; 2022) for more explanations on baselines. The majority of the experimental analysis results include the models **SASRec_AC** (Xin et al., 2020), **SNQN** and **SA2C** (Xin et al., 2022) which are closely related to our method. All the models presented in Figures 3 and 4 use the SASRec model as the base model and use the actor-critic framework outlined in Figure 2. The baselines **SNQN** performs a naive negative sampling, and **SA2C** includes the advantage estimations to re-weight the Q-values.

**Evaluation protocols.** We adopt cross-validation to evaluate the performance of the proposed methods using the same data split proposed in (Xin et al., 2022). Every experiment is conducted using 5 random seeds, and the average performance of the top 5 best performing checkpoints is reported and the visualization plots demonstrate the training progression across all 5 seeds, including variance. The recommendation quality is measured with two metrics: **Hit Ratio (HR)** and **Normalized Discounted Cumulative Gain (NDCG)**. HR@$k$ is a recall-based metric, measuring whether the ground-truth item is in the top-k positions of the recommendation list. NDCG is a rank sensitive metric which assign higher scores to top positions in the recommendation list. We focus on the two extremes of Top-5 and Top-20 to compare all methods.

### 4.2 MAIN RESULTS

We show the performance of recommendations on RetailRocket and RC15 in Table 1. Table 2 showcases the results on Yelp (Yelp, 2021). We also show the relative improvement compared with best baseline models. To ensure reproducibility and fairness, we re-executed the best-performing baseline models proposed in (Xin et al., 2020; 2022). The results are derived from an average of five runs. Due to space constraints, only the average results are presented in the main paper; for additional statistical information, including error bars, please consult Appendix A.3.

Table 1: Top-$k$ ($k$ = 5, 10, 20) performance comparison of different models on **RetailRocket** and **RC15** on the task of **Purchase** prediction.

| Datasets | Metric@k | SASRec-AC | SA2C | CL4SRec | SASRec-CDARL | CP4Rec-CDARL | FMLPRec* | SASRec-CO | SASRec-CCQL | Improv |
|---|---|---|---|---|---|---|---|---|---|---|
| | HR@5 | 0.606 | 0.612 | 0.518 | 0.578 | 0.581 | 0.587 | 0.611 | **0.613** | 0.2% |
| | HR@10 | 0.651 | 0.660 | 0.560 | 0.631 | 0.639 | 0.631 | 0.666 | **0.676** | 3.8% |
| RetailRocket | HR@20 | 0.687 | 0.689 | 0.598 | 0.678 | 0.684 | 0.669 | 0.706 | **0.720** | 4.5% |
| | NDCG@5 | 0.515 | 0.512 | 0.443 | 0.479 | 0.479 | 0.504 | 0.513 | **0.517** | 0.4% |
| | NDCG@10 | 0.531 | 0.527 | 0.457 | 0.498 | 0.498 | 0.518 | 0.532 | **0.533** | 0.4% |
| | NDCG@20 | 0.549 | 0.554 | 0.466 | 0.508 | 0.508 | 0.528 | 0.542 | **0.569** | 2.7% |
| | HR@5 | 0.444 | 0.470 | 0.399 | 0.452 | 0.444 | 0.439 | 0.419 | **0.496** | 5.5% |
| | HR@10 | 0.562 | 0.575 | 0.516 | 0.566 | 0.564 | 0.542 | 0.536 | **0.620** | 7.8% |
| RC15 | HR@20 | 0.643 | 0.664 | 0.601 | 0.655 | 0.652 | 0.625 | 0.622 | **0.712** | 7.2% |
| | NDCG@5 | 0.321 | 0.338 | 0.285 | 0.317 | 0.311 | 0.320 | 0.298 | **0.356** | 5.3% |
| | NDCG@10 | 0.359 | 0.372 | 0.323 | 0.355 | 0.350 | 0.354 | 0.336 | **0.397** | 6.7% |
| | NDCG@20 | 0.380 | 0.395 | 0.345 | 0.378 | 0.372 | 0.375 | 0.358 | **0.419** | 6.1% |

Table 2: Top-$k$ ($k$ = 5, 10, 20) performance comparison of different models on **Yelp**.

| Model | Reward@20 | Purchase | | | | | | Click | | | | | |
|---|---|---|---|---|---|---|---|---|---|---|---|---|---|
| | | HR@5 | NG@5 | HR@10 | NG@10 | HR@20 | NG@20 | HR@5 | NG@5 | HR@10 | NG@10 | HR@20 | NG@20 |
| NextItNet (Yuan et al., 2019) | 392 | 0.342 | 0.310 | 0.363 | 0.317 | 0.423 | 0.332 | 0.475 | 0.412 | 0.516 | 0.426 | 0.572 | 0.440 |
| NextItNet-AC (Xin et al., 2020) | 151 | 0.127 | 0.094 | 0.151 | 0.101 | 0.205 | 0.116 | 0.087 | 0.067 | 0.110 | 0.074 | 0.153 | 0.085 |
| Caser (Tang & Wang, 2018) | 421 | 0.396 | 0.362 | 0.427 | 0.372 | 0.466 | 0.382 | 0.485 | 0.407 | 0.537 | 0.424 | 0.581 | 0.436 |
| GRU-AC (Xin et al., 2020) | 397 | 0.439 | 0.358 | 0.488 | 0.374 | 0.537 | 0.387 | 0.289 | 0.224 | 0.341 | 0.241 | 0.390 | 0.254 |
| SASRec (Xin et al., 2020) | 436 | 0.417 | 0.360 | 0.456 | 0.373 | 0.487 | 0.381 | 0.553 | 0.469 | 0.596 | 0.484 | 0.633 | 0.493 |
| SASRec-AC (Xin et al., 2020) | 449 | 0.403 | 0.353 | 0.457 | 0.370 | 0.500 | 0.381 | 0.529 | 0.455 | 0.598 | 0.477 | 0.649 | 0.490 |
| SNQN (Xin et al., 2022) | 417 | 0.388 | 0.351 | 0.415 | 0.359 | 0.448 | 0.367 | 0.545 | 0.466 | 0.584 | 0.479 | 0.631 | 0.491 |
| SA2C (Xin et al., 2022) | 404 | 0.409 | 0.382 | 0.421 | 0.376 | 0.450 | 0.393 | 0.547 | 0.484 | 0.577 | 0.485 | 0.611 | 0.503 |
| CL4Rec (Xie et al., 2022) | 457 | 0.384 | 0.284 | 0.463 | 0.310 | 0.506 | 0.321 | 0.539 | 0.421 | 0.614 | 0.445 | 0.670 | 0.460 |
| SASRec-CO (Ours) | 450 | 0.421 | 0.362 | 0.452 | 0.372 | 0.473 | 0.378 | 0.537 | 0.447 | 0.595 | 0.466 | 0.653 | 0.506 |
| **SASRec-CCQL (Ours)** | **508** | **0.457** | **0.389** | **0.531** | **0.384** | **0.572** | **0.394** | **0.582** | **0.496** | **0.641** | **0.488** | **0.690** | **0.510** |
| Improvment | 11.2% | 4.1% | 1.8% | 8.8% | 2.1% | 6.5% | 0.2% | 5.2% | 2.5% | 4.4% | 0.6% | 3.0% | 1.4% |

Our method outperforms all baseline models across all examined datasets in all metrics. The combination of a sequential contrastive learning objective with improvements to the negative sampling methodology consistently yields improved performance over the baselines.

## 4.3 ROBUSTNESS STUDIES

In this section, we show the robustness of our proposed method **SASRec-CO**. It is SASRec-AC with the added contrastive objective. There is no negative action sampling and the contrastive objective is applied solely across batches of data as positive and negative items. **SASRec-CCQL** adopts negative *action* sampling and employs both the contrastive and conservative objectives outlined in Eq. 4. Our empirical analysis underscores the need to monitor training progress for RL-based models to detect instabilities that could impair model performance in online deployment. An observable trend throughout our experiments is that the baseline methods SNQN and SA2C initially attain high accuracy, but their performance rapidly deteriorates as Q-learning diverges. We advocate for the reporting training dynamics alongside tabular results when reporting recommender model performances.

All baselines in Figure 3 employ a negative sampling set to 10, with the exception of SASRec-CO. The number of negative samples to be selected per training batch depends on the length of the sequences in the data. In the context of executing the baseline code provided by (Xin et al., 2022), the original parameters were used to run the methods. For the baseline SA2C, the smoothing parameter, which is responsible for applying the off-policy correction, was initially set to 0, effectively disabling this correction term. Therefore the baseline SA2C does not include the correction term. However it does involve a double optimization strategy as discussed in (Xin et al., 2022) and the usage of advantage estimation which does provide improvement over SNQN.

Nevertheless, our approach exhibits robustness; even though the learning process for RetailRocket Figure 3 is slower, the policy remains stable on the long run and surpasses performance across all baselines. A similar trend is apparent in the RC15 Figure 4 experiments, where the number of negative samples does not detrimentally impact the performance, and training stability is maintained. Conversely, for the baseline methods SNQN and SA2C, a performance decline or divergence is observed throughout the training process, which is further amplified by an increase in the negative samples.

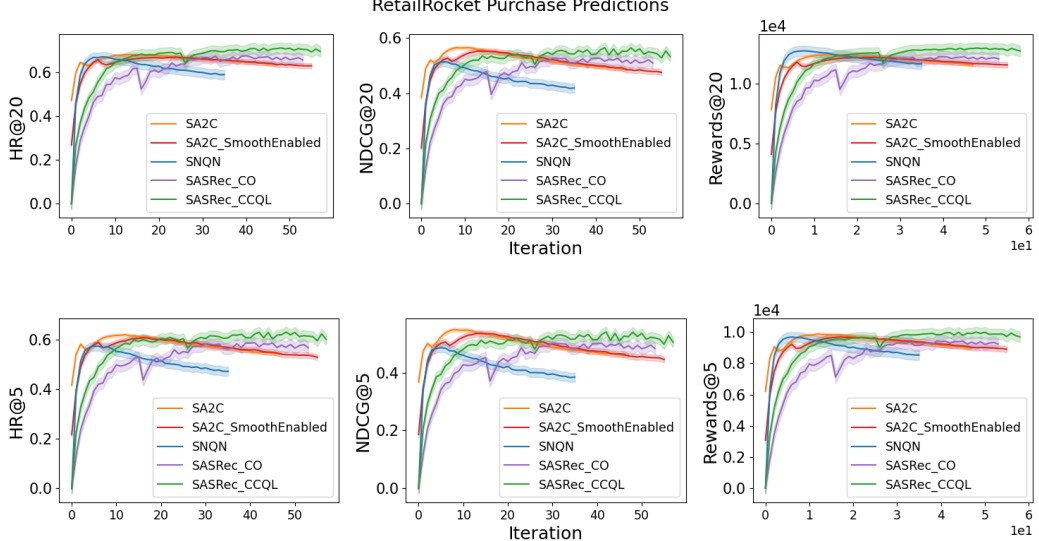

Figure 3: Our method SASRec-CCQL outperforms other approaches in predicting purchases for both Top-20 and Top-5 recommendations.

## 4.4 ABLATION STUDIES

Ablation studies on different components are show in table 3 and 4 for RetailRocket and RC15 respectively. These ablation studies show with our added conservative and contrastive approaches achieve the best results in almost all metrics. We also perform ablation study on discount factor and effect of RL, as well as overestimation bias, please refer to Appendix A.1 for more details.

Table 3: Top-$k$ ($k$ = 5, 10, 20) ablation study on **RetailRocket**.

| Model | Reward@20 | Purchase | | | | | | Click | | | | | |
|---|---|---|---|---|---|---|---|---|---|---|---|---|---|
| | | HR@5 | NG@5 | HR@10 | NG@10 | HR@20 | NG@20 | HR@5 | NG@5 | HR@10 | NG@10 | HR@20 | NG@20 |
| SASRec | 12,240 | 0.586 | 0.493 | 0.642 | 0.512 | 0.681 | 0.521 | 0.263 | 0.201 | 0.317 | 0.218 | 0.364 | 0.230 |
| SASRec-AC | 12,448 | 0.606 | 0.525 | 0.651 | 0.533 | 0.687 | 0.549 | 0.270 | 0.209 | 0.323 | 0.226 | 0.372 | 0.239 |
| SASRec-CO | 12,897 | 0.611 | 0.513 | 0.666 | 0.532 | 0.706 | 0.542 | **0.280** | 0.214 | 0.335 | 0.230 | 0.387 | **0.245** |
| **SASRec-CCQL** | **12,987** | **0.613** | **0.517** | **0.676** | **0.533** | **0.720** | **0.569** | **0.280** | **0.220** | **0.336** | **0.236** | 0.387 | **0.245** |

Table 4: Top-$k$ ($k$ = 5, 10, 20) ablation study on **RC15**.

| Model | Reward@20 | Purchase | | | | | | Click | | | | | |
|---|---|---|---|---|---|---|---|---|---|---|---|---|---|
| | | HR@5 | NG@5 | HR@10 | NG@10 | HR@20 | NG@20 | HR@5 | NG@5 | HR@10 | NG@10 | HR@20 | NG@20 |
| SASRec | 13,404 | 0.391 | 0.273 | 0.494 | 0.307 | 0.585 | 0.330 | 0.322 | 0.224 | 0.416 | 0.255 | 0.492 | 0.274 |
| SASRec-AC | 14,010 | 0.444 | 0.321 | 0.562 | 0.359 | 0.643 | 0.380 | 0.343 | 0.239 | **0.439** | 0.338 | 0.517 | 0.290 |
| SASRec-CO | 13,782 | 0.419 | 0.298 | 0.536 | 0.336 | 0.622 | 0.358 | 0.332 | 0.226 | 0.428 | 0.262 | 0.506 | 0.278 |
| **SASRec-CCQL** | **14,311** | **0.496** | **0.356** | **0.620** | **0.397** | **0.712** | **0.419** | **0.348** | **0.239** | 0.427 | **0.264** | **0.508** | **0.291** |

## 4.5 DISCUSSION ON LIMITATIONS

While demonstrating the efficacy of an offline RL solution such as CQL, it is crucial to acknowledge that it is not universally optimal. For instance, in online RL scenarios that entail an agent learning through interaction, the effectiveness of CQL may diminish. Furthermore, the conservative nature of CQL can potentially result in the underestimation of Q-values, giving rise to overly cautious policies that may not always align with the requirements of a recommender model.

Moreover, the current publicly available datasets, characterized by brief user interactions and simplistic reward functions, impose limitations on the full potential of RL. A promising avenue for future research lies in the development of recommender system benchmarks specifically tailored

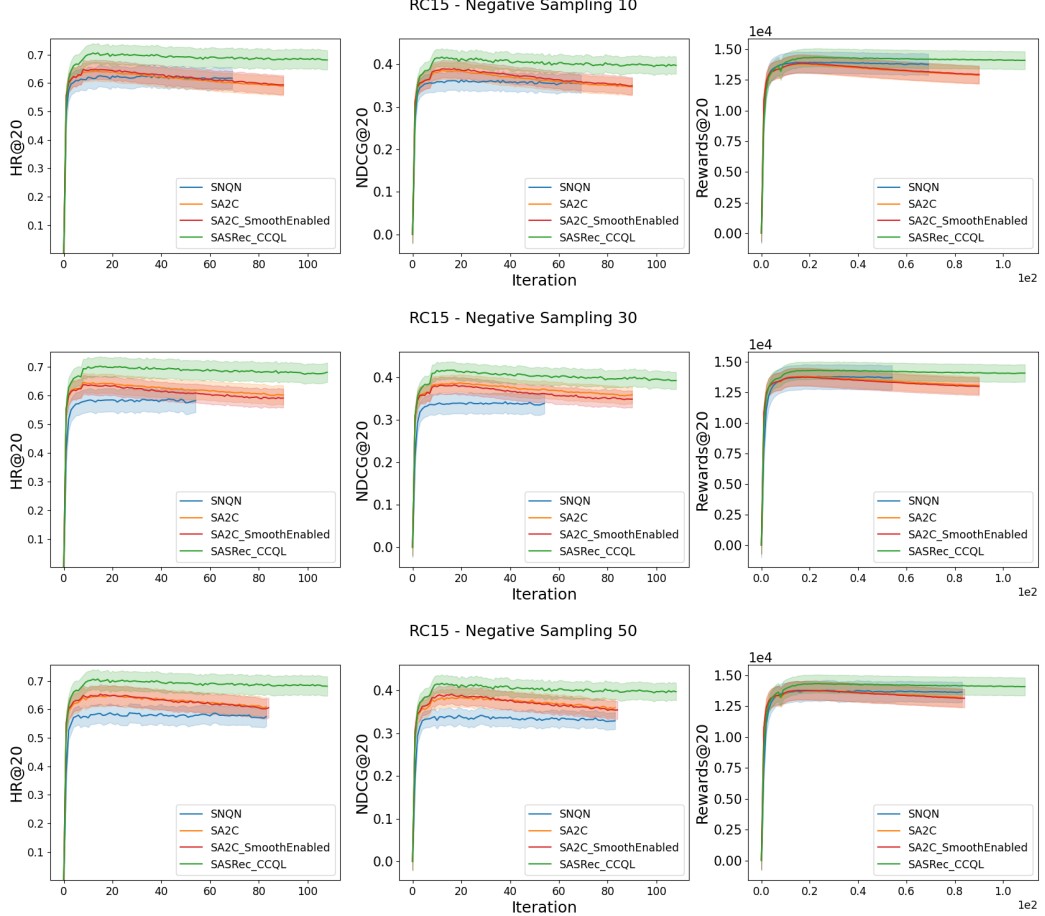

Figure 4: Purchase predictions comparisons on **Top-20** for varying negative samplings. These results demonstrate higher performance is achieved and remains stable with increasing negative samples, unlike baseline methods SNQN and SA2C, which exhibit performance decline and divergence.

for RL, with the objective of gaining a deeper understanding of user preferences and enhancing personalization capabilities.

## 5 CONCLUSION

Our research unveils novel insights into the effectiveness of integrating contrastive learning into recommender systems. This approach offers richer representations of states and actions, thereby augmenting the learning potential of the Q-function within the contrastive embedding space. Consequently, it enables a more precise differentiation between states and actions.

Moreover, the conservative nature of Q-learning introduces a valuable equilibrium, preventing the overestimation of Q-values that could otherwise potentially lead to sub-optimal policies. This adjustment in Q-learning safeguards against excessively optimistic assumptions regarding the rewards associated with certain actions.

Additionally, we discovered that the incorporation of negative action sampling significantly enhances the overall performance of the model and ensures stability in RL training. Although not revolutionary in nature, this amalgamation constitutes a substantial contribution to the field, representing a meaningful advancement in our understanding of reinforcement learning.

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
