## A APPENDIX FOR ADDRESSING CHALLENGES IN REINFORCEMENT LEARNING FOR RECOMMENDER SYSTEMS WITH CONSERVATIVE OBJECTIVES

### A.1 ABLATION STUDY

**Discount Factor and Effect of RL:** In reinforcement learning, the discount factor $\gamma$ is a parameter that determines the importance of future rewards. When we set the $\gamma$ to 0, we expect the agent to only care about the immediate reward and not consider future rewards at all. It becomes a greedy agent, focusing only on maximizing the immediate reward. In other words, the agent will become myopic or short-sighted. On the other hand, if we set $\gamma$ to 0.99 (close to 1), the agent will put more emphasis on future rewards in its decision-making process. This encourages more exploration and a more far-sighted policy. The agent is driven to strike a balance between immediate and future rewards. In a typical recommender system scenario, a model that contemplates long-term user preferences is usually desirable. In this section, we examine the impact of this parameter on the system's performance, which could indicate the potential benefits of integrating the model with Reinforcement Learning (RL). While our datasets at hand are relatively short-horizon (compared to control tasks in robotics), characterized by brief interaction sequences per user, the full potential of RL isn't entirely manifested since typically, RL considers a much longer horizon by factoring in higher discount rates. This approach enables the model to emphasize the significance of future rewards, promoting a far-sighted perspective in decision-making processes. However, we do witness a noticeable enhancement when incorporating the RL component, indicating a promising direction for further exploration and optimization. We hypothesize that by employing our framework on a dataset more suited to reward-oriented learning, we could witness a more significant advantage from the application of RL. This could unlock more robust policies and performance improvements, further highlighting the potential of RL in such contexts.

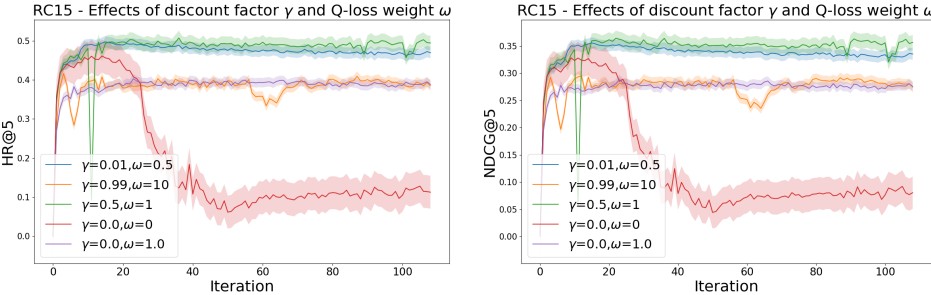

Figure 5: Effect of discount factor $\gamma$ and $\omega$ which scales the magnitude of the Q-loss on Top-5 purchase predictions on the dataset RC15. When $\gamma$ is set to 0, the agent will become myopic, caring only about the immediate rewards. When $\gamma$ is set closer to 1 (i.e. 0.99) the policy prioritizes long-term rewards over immediate rewards i.e. becomes long-sighted. This has overall effect on the performance of the system as whole, where we observe optimal performance and stability with $\gamma = 0.5$ & $w = 1.0$.

**Overestimation Bias:** Overestimation bias in Q-learning refers to the consistent over-evaluation of the expected reward of specific actions by the Q-function (action-value function), resulting in less than optimal policy decisions. This phenomenon can potentially be visualized in several ways. One of the primary challenges in offline RL revolves around the problem of distributional shift. From the agent's perspective, acquiring useful abilities requires divergence from the patterns observed in the dataset, which necessitates the ability to make counterfactual predictions, that is, speculating about the results of scenarios not represented in the data. Nonetheless, reliable counterfactual predictions become challenging for decisions that significantly diverge from the dataset's behavior. Due to the conventional update method in RL algorithms, for instance, Q-learning which involves querying the Q-function at out-of-distribution inputs to calculate the bootstrapping target during the training process. As a result, standard off-policy deep RL algorithms often tend to inflate the values of such unseen outcomes i.e. negative actions. This causes a shift away from the dataset towards what seems

like a promising result, but actually leads to failure. In order to successfully navigating the trade-off between learning from the offline data and controlling overestimation bias, we prefer Q-values that disentangle the distinction between the two (seen and unseen samples) more discernible.

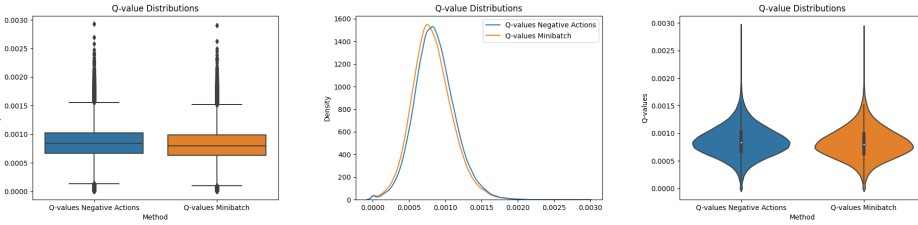

Figure 6: The Q-values are set to the same initial value across all methods. Initially the mini-batch and negative actions are treated the same since no learning has taken place.

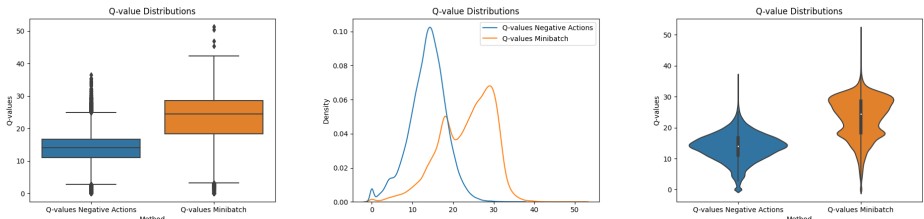

Figure 7: CQL final Q-values for the final policy. Ideally we prefer to see more distinction between evaluating the Q-function on the mini-batch samples vs negative actions. For the Q-value distributions, the x-axis represents the Q-values and we prefer a distribution more shifted towards positive values for the mini-batch and a narrow, lower Q-values region for the low-reward negative actions.

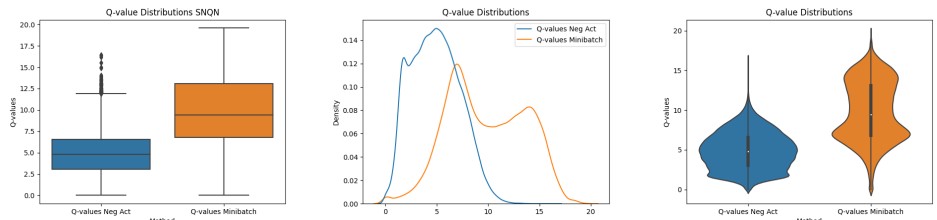

Figure 8: SNQN final Q-values. In these plots, we observe more overlap between the evaluation of the Q-function on the mini-batch vs negative actions. This can be undesirable and has lead to reduced overall performance of the final policy.

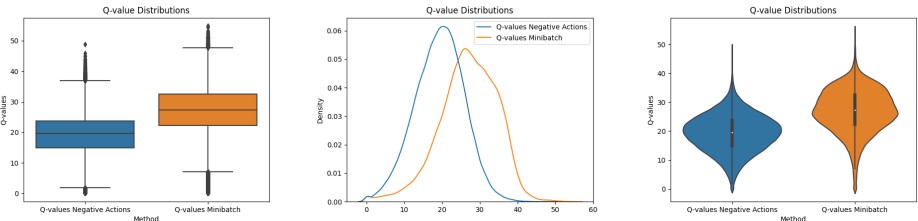

Figure 9: SA2C final Q-values. We observe similar trend as the other baseline SNQN, however there is more overlap between the Q-value distributions as in this case the policy has diverged, leading to lower performance.

## A.2 ADDITIONAL RESULTS RC15

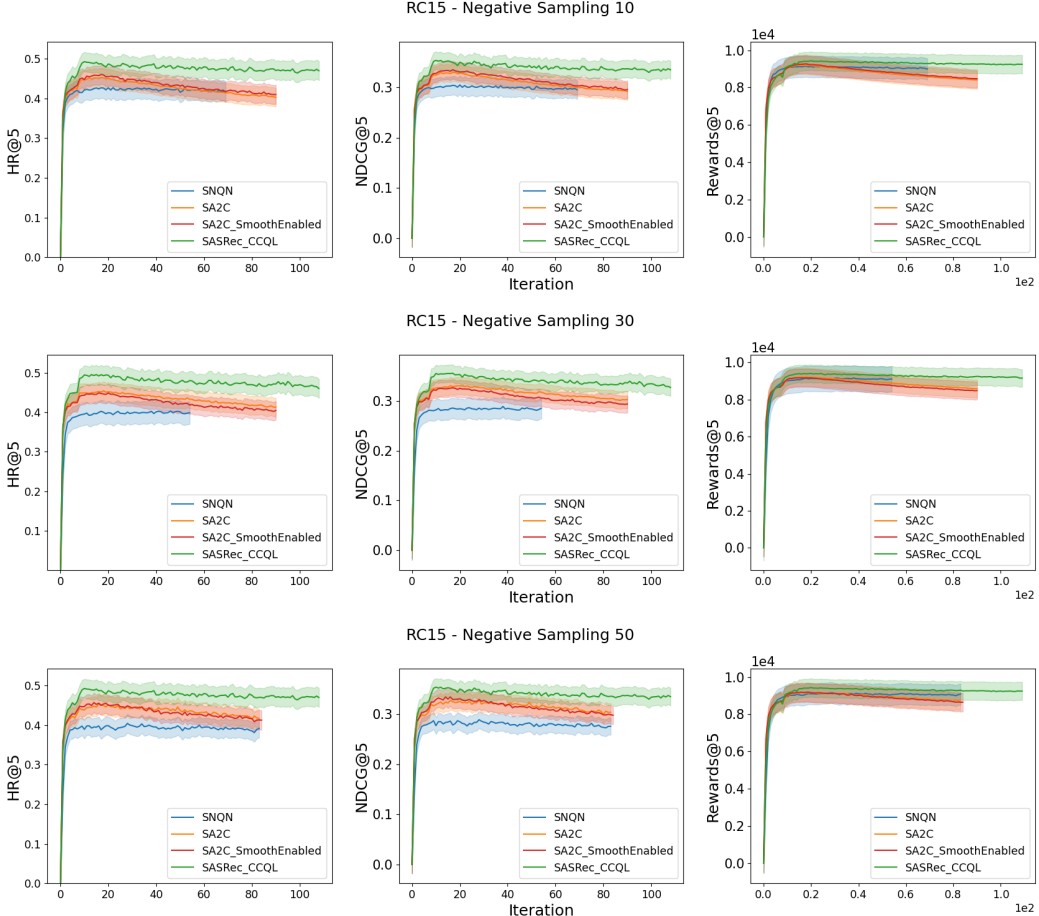

Figure 10: Purchase predictions comparisons on **Top-5** for varying negative samplings. As we increase the rate of negative samples during training, we observe performance drop in our baseline SNQN and divergence with SA2C and SA2C with smoothing i.e. off-policy correction enabled.

## A.3 RESULTS STATISTICS

Table 5: Top-$k$ ($k$ = 5, 10, 20) performance comparison of different models on **RC15** including mean and standard deviation errors, averaged across 10 seeds.

| Model | Reward@20 | Purchase | | | | | | Click | | | | | |
|---|---|---|---|---|---|---|---|---|---|---|---|---|---|
| | | HR@5 | NG@5 | HR@10 | NG@10 | HR@20 | NG@20 | HR@5 | NG@5 | HR@10 | NG@10 | HR@20 | NG@20 |
| SASRec | $13,181 \pm 14$ | $0.379 \pm 0.006$ | $0.271 \pm 0.004$ | $0.482 \pm 0.004$ | $0.304 \pm 0.003$ | $0.564 \pm 0.003$ | $0.325 \pm 0.003$ | $0.318 \pm 0.001$ | $0.222 \pm 0.001$ | $0.41 \pm 0.001$ | $0.252 \pm 0.001$ | $0.487 \pm 0.001$ | $0.271 \pm 0.001$ |
| SASRec-AC | $13,693 \pm 28$ | $0.393 \pm 0.004$ | $0.278 \pm 0.003$ | $0.497 \pm 0.004$ | $0.312 \pm 0.002$ | $0.583 \pm 0.006$ | $0.334 \pm 0.002$ | $0.333 \pm 0.001$ | $0.232 \pm 0.001$ | $\mathbf{0.428 \pm 0.001}$ | $0.263 \pm 0.001$ | $0.507 \pm 0.001$ | $0.283 \pm 0.001$ |
| SASRec-CO | $13,701 \pm 25$ | $0.392 \pm 0.003$ | $0.279 \pm 0.002$ | $0.498 \pm 0.004$ | $0.313 \pm 0.002$ | $0.584 \pm 0.005$ | $0.335 \pm 0.002$ | $0.333 \pm 0.001$ | $\mathbf{0.233 \pm 0.001}$ | $0.427 \pm 0.001$ | $\mathbf{0.264 \pm 0.001}$ | $0.507 \pm 0.001$ | $0.283 \pm 0.001$ |
| SASRec-CCQL | $\mathbf{14,187 \pm 57}$ | $\mathbf{0.473 \pm 0.006}$ | $\mathbf{0.338 \pm 0.004}$ | $\mathbf{0.596 \pm 0.006}$ | $\mathbf{0.377 \pm 0.004}$ | $\mathbf{0.688 \pm 0.005}$ | $\mathbf{0.401 \pm 0.004}$ | $\mathbf{0.348 \pm 0.001}$ | $0.227 \pm 0.001$ | $0.426 \pm 0.001$ | $0.259 \pm 0.001$ | $\mathbf{0.508 \pm 0.002}$ | $\mathbf{0.291 \pm 0.001}$ |

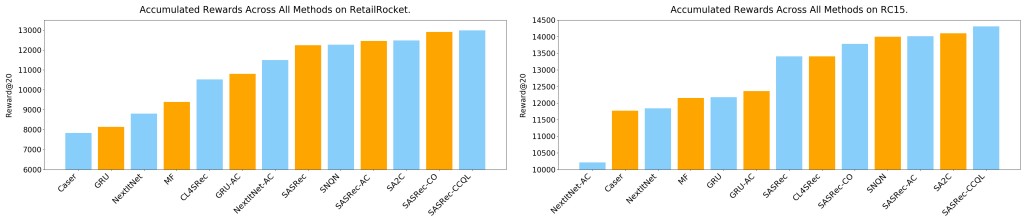

Figure 11: Comaprison of accummulated rewards across all methods.

## A.4 ADDITIONAL DATASETS

| Model | Click | | | | | |
|---|---|---|---|---|---|---|
| | HR@5 | NG@5 | HR@10 | NG@10 | HR@20 | NG@20 |
| GRU | 0.390 | 0.259 | 0.515 | 0.298 | 0.578 | 0.314 |
| Caser | 0.265 | 0.206 | 0.328 | 0.226 | 0.484 | 0.265 |
| SASRec | 0.351 | 0.240 | 0.421 | 0.263 | 0.558 | 0.298 |
| **SASRec-CO** | **0.453** | **0.350** | **0.593** | **0.402** | **0.687** | **0.426** |
| CL4Rec | 0.328 | 0.234 | 0.394 | 0.255 | 0.527 | 0.289 |

Table 6: Top-$k$ performance comparison of different models ($k = 5, 10, 20$) on **MovieLens**.

| Model | Click | | | | | |
|---|---|---|---|---|---|---|
| | HR@5 | NG@5 | HR@10 | NG@10 | HR@20 | NG@20 |
| GRU | 0.031 | 0.023 | 0.046 | 0.028 | 0.093 | 0.040 |
| Caser | 0.015 | 0.015 | 0.031 | 0.020 | 0.031 | 0.020 |
| SASRec | 0.023 | 0.018 | 0.054 | 0.027 | 0.070 | 0.032 |
| **SASRec-CO** | **0.046** | **0.032** | **0.078** | **0.041** | **0.109** | **0.048** |
| CL4Rec | 0.023 | 0.015 | 0.039 | 0.020 | 0.046 | 0.022 |

Table 7: Top-$k$ performance comparison of different models ($k = 5, 10, 20$) on **AmazonFood**.

## A.5 HYPERPARAMETERS

| Hyperparameter | Initial Value | Tuning Range |
|---|---|---|
| Batch_size | 256 | 32 to 128 |
| Hidden_size | 64 | 32 to 128 |
| Learning Rate | 0.001 | 1e-5 to 0.1 |
| Discount ($\gamma$) | 0.5 | 0.001 to 0.999 |
| Contrastive Loss | InfoNCECosine | N/A |
| Augmentation | Permutation | N/A |
| Negative Reward | -1.0 | -5 to 0 |
| Negative Samples | 10 | 10 to 50 |
| CQL Temperature | 1.0 | 0.1 to 5 |
| CQL Min Q Weight | 0.1 | 0.001 to 5.0 |
| Q Loss Weight | 0.5 | 0.1 to 2.0 |

Table 8: Hyperparameters for SASRec-CCQL

| Hyperparameter | Initial Value | Tuning Range |
|---|---|---|
| Batch_size | 256 | 32 to 128 |
| Hidden_size | 64 | 32 to 128 |
| Learning Rate | 0.01 | 1e-5 to 0.1 |
| Discount ($\gamma$) | 0.1 | 0.001 to 0.999 |
| Contrastive Loss | InfoNCE | N/A |
| Augmentation | Permutation | N/A |
| Negative Reward | -1.0 | -5 to 0 |

Table 9: Hyperparameters for SASRec-CO