# OpenReview forum: "Addressing Challenges in Reinforcement Learning for Recommender Systems with Conservative Objectives"
_ICLR.cc/2024/Conference — ICLR 2024 Conference Withdrawn Submission_

### Official Review · Reviewer_EFLR · 2023-10-26

**Soundness:** 2 fair
**Presentation:** 3 good
**Contribution:** 2 fair
**Rating:** 5
**Confidence:** 5

**Summary:**

This paper study the topic of RL-based recommender systems. The paper claims the instability problem of previous RL-based RS algorithms, and propose to use the conservative Q learning algorithm to improve the training stability. The paper further uses contrastive learning to learn better representation of samples. Experiments on five public datasets show that SARSRec-CCQL outperforms SA2C, SNQN and CL4SREC.

**Strengths:**

The paper is well written and easy to follow. 2. The instability of RL-based RS algorithm is a good problem to study. 3. The paper provides the training curve of the proposed algorithm and baselines, and provide the ablation study. 4.The performance improvement is good.

**Weaknesses:**

1.The paper aims to study the instability problem of RL-based RS algorithms, thus the conservative Q learning algorithm is proposed. However, I think the novelty of the paper is not enough. I suggest that the paper does more analysis on the instability of training SARSRec and how the CQL algorithm improves the stability in recommendations.

2.The paper claims that it can tackle the problem of contrastive learning in recommendation by using a conservative objective, which is a minor point. I think the conservative objective is a general method to improve the training stability for offline RL algorithms. However，the paper does not discuss it.

3.The paper does not provide the ablation study on the conservative Q-learning loss, such as the effect of the value of alpha in Eq.(2).

**Questions:**

See the above part.

---

### Official Review · Reviewer_zZt2 · 2023-10-30

**Soundness:** 2 fair
**Presentation:** 3 good
**Contribution:** 3 good
**Rating:** 5
**Confidence:** 4

**Summary:**

The paper explores the integration of reinforcement learning into attention-based sequential recommendation models, aiming to enhance personalization by incorporating user feedback as rewards. The authors focus on the critical problem of training stability in RL algorithms within a sequential learning context. They introduce innovative solutions, featuring contrastive learning objectives for enhanced representation learning and Conservative Q-learning to improve policy and value learning while mitigating off-policy training issues. Their approach is aimed at addressing data distribution challenges, particularly in negative action sampling, reducing positive action biases and value overestimation. This work stands out for its emphasis on ensuring the stability and robustness of RL-based recommender systems, offering valuable insights into improving training methodologies for real-world deployment.

The paper is well-structured and easy to read. The logic of designing the solution is transparent and supports the main claims of the work. The text is accompanied with clear graphical representations that simplify understanding of the proposed approach. However, there are several major issues that prevent connecting the conclusions of the paper to the initial goals.

**Strengths:**

- the paper highlights the important problem of training instabilities in RL algorithms for sequential recommendations tasks and offers a solution to that
- the design of the solution is novel and interesting and can be considered a valuable contribution
- the authors explore an important link between training instabilities and proper learning of underlying data distribution

**Weaknesses:**

- the work lacks appropriate baseline comparison to isolate RL component from other architectural modifications
- the paper suffers from methodological issues w.r.t sequential learning and recommendations quality evaluation, no source code is provided
- some of the benefits of using RL in recsys are overstated and lack sufficient evidence supporting the claims

**Questions:**

### **Main concerns**
My main concern is that the provided comparison with baselines doesn't help to create a convincing picture of the superiority of the proposed RL-based approach over standard attention-based sequential learning models. The key limitation of the work is that non-RL baseline algorithms significantly differ from the proposed solution. There are several aspects in which this difference is crucial in the view of understanding the true contribution of RL. First of all, the loss function. The proposed solution (as well as several RL baselines that the authors use) utilizes full cross-entropy (CE) loss (the actor part), while the original SASRec model uses binary cross-entropy (BCE) loss. This change alone provides a significant difference in the learning of proper data distribution. BCE serves is a very rough approximation for learning underlying data structure, it significantly depends on the negative sampling strategy. In the case of SASRec it's based on a uniform sampling with a single negative example being drawn for each positive interaction. This is a very strict limitation on data exploration part. So at least two modifications must be introduced into the non-RL SASRec model before comparing to the RL models:
1) increasing the number of negative samples in BCE,
2) switching from BCE to CE loss.

These changes are self-evident and must be easy to implement considering that the architecture of the proposed solution already has the necessary components.  Nevertheless, the work by [Klenitskiy and Vasilev 2023] may also be helpful in the context of the current paper as the authors perform a direct comparison of SASRec with different loss functions that include BCE, CE and subsampled CE. Similar results regarding the advantages of CE loss were obtained in e.g., [Frolov et al. 2023]. Some work related to improving negative sampling in BCE was also done by [Petrov and Macdonald 2023]. Most importantly, these works demonstrate a significant improvement over the baseline SASRec model and therefore the comparison with the modified version of SASRec (not just its basic implementation) is absolutely necessary.

Another concern is related to identifying and quantifying the training instability issue. It also connects to the previous concern as, for example, the instability in the case of SASRec can arise from the reinforced bias against true negative samples induced by BCE loss (which is only an approximation for learning data distribution as stated above). To make comparison more reliable and informative, the learning curves for the SASRec model with full CE loss (at least) are necessary.

Moreover, the figures provided to demonstrate the effect of training instability are truncated to only the first 50 iterations of training. Considering the slow-paced learning of the proposed model, the graph is not convincing. What happens at e.g., 200 iterations (200 iterations is a moderate number that is met across many models)? Does the method still behave in a stable fashion? From the current figure it can only be stated that it's more stable only within the first 50 iterations. But if it degrades afterwards than the method provides no advantage, it only postpones the problem till later time. In addition to that, the quality of the proposed approach does not seem to be higher. It's hard to see from the graph on Figure 3, though, some horizontal lines emphasizing the maximum achieved metric value would help visual analysis. Anyway, at least in some figures, other methods seem to achieve the highest quality faster (e.g., for NDCG metric), which will completely diminish advantages of the proposed approach if it also suffers from instabilities but at a later stage (higher number of iterations).

The third major concern is related to the overall methodology of experiments. First of all, it's not clear what metric was used to validate the models? Connecting it to the previous point, if the target metric was HR and the model shows higher HR at the cost of lower NDCG, it may signify an overfitting problem of the approach. Secondly, in sequential learning it's critical to avoid data leakage. A widespread problem with many works including SASRec is that they use leave-last-out scheme to hide target items for test and validation, which inevitably leads to recommendations-from-future type of leaks. See e.g. [Ji at el 2023] for more details on proper splitting. In addition to that, while the authors provide a link to the source code repository, the repository is empty, which raises reproducibility concerns.

The current work seems to have an even more significant leakage problem as target items are held out randomly, not even according to the sequential order. This makes the results look weak. Another issue related to the evaluation setup is the target items subsampling used to calculate the metrics, in which the ground truth item is ranked agains only a small subsample of negative examples instead of entire catalog. This scheme leads to various estimation biases (see [Cañamares and Castells 2020], [Krichene and Rendle 2020]) and should be generally avoided unless one really works with huge item catalogs of million-size.

Related to the metrics, it is mentioned in the text that learning an optimal policy allows balancing the trade-off between exploration and exploitation. Better exploration vs exploitation will then result in more diverse recommendations (rather than just focusing on more popular recommendations). However, no evidence for this is provided in the results. Some simple metrics like item catalog coverage (i.e., fraction of the number of unique recommended items to the entire catalog size) would help to provide the necessary ground for the claim.

Please also add a table with the overall statistics for all datasets. Currently, only the RetailRocket dataset is provided with such information.

### **Other concerns**
I also have a few minor conceptual issues. As stated in the text, the general formulation of the RL problem involves incorporating direct user feedback in the form of rewards, which, in my opinion, leads to the "chicken and egg problem". The quality of recommendations would directly depend on the quality of the environment and reward functions, which serve only as an approximation. If the reward function is bad, how can one hope to suddenly have an accurate recommendation model? But if the environment and the reward function are very accurate, why would you need RL on top of it? Why not just use the environment and rewards directly to generate recommendations? Reading the text feels like RL algorithms have already proved their superiority, but no sufficient evidence is provided due to the lack of fair comparison with strong and finely tuned non-RL baselines.

Probably the most standing out promise of RL, which is also mentioned by the authors, is the promise of maximizing long-term user satisfaction. This is a frequent and plausible claim, although actual evidence of that is still an open question. It's not even clear how to measure the long-term effects in offline data. The statement on the long-term effects creates a false impression that the problem is already solved and that there's a scientific consensus on superiority of RL in that regard.

The authors report monitoring the of training progression as an individual contribution of the work.  No doubt this monitoring is useful in diagnosing and comparing models, but it cannot be a contribution, it's not something new or unused in practice.

Please also note that the term "stability" has another meaning in the realm of recommender systems related to the rate of change of recommendations when the model is updated with new information [Olaleke et al. 2021]. This "recommendations instability" is very different from the "training instability". A clear distinction between the two somewhere in the introduction would help the reader to avoid confusion.

There are several issues with constantly changing notation, some letters' font interchangeably becomes straight or italic/caligraphic back and forth. Please fix the notation and use it consistently across the text.

The first appearance of the Q-value term in the text is not explained, leaving an unprepared reader with the necessity to lookup the literature before proceeding. Adding a few words explaining what Q-value is would improve text readability.

In page 4, paragraph before section 3.2, there seems to be a confusion between "off policy" and "offline" training.

### **References**
Klenitskiy, A. and Vasilev, A., 2023, September. Turning Dross Into Gold Loss: is BERT4Rec really better than SASRec?. In _Proceedings of the 17th ACM Conference on Recommender Systems_ (pp. 1120-1125).

Frolov E., Bashaeva L., Mirvakhabova L., Oseledets I. Hyperbolic Embeddings in Sequential Self-Attention for Improved Next-Item Recommendations. Under review at https://openreview.net/forum?id=0TZs6WOs16.

Petrov, A.V. and Macdonald, C., 2023, September. gSASRec: Reducing Overconfidence in Sequential Recommendation Trained with Negative Sampling. In _Proceedings of the 17th ACM Conference on Recommender Systems_ (pp. 116-128).

Ji Y, Sun A, Zhang J, Li C. A critical study on data leakage in recommender system offline evaluation. ACM Transactions on Information Systems. 2023 Feb 7;41(3):1-27.

Cañamares, R., and  Castells, P. "On target item sampling in offline recommender system evaluation." In Fourteenth ACM Conference on Recommender Systems, pp. 259-268. 2020

Krichene, W, and Rendle, S. "On sampled metrics for item recommendation." In Proceedings of the 26th ACM SIGKDD international conference on knowledge discovery & data mining, pp. 1748-1757. 2020.

Olaleke, O., Oseledets, I. and Frolov, E., 2021, June. Dynamic modeling of user preferences for stable recommendations. In Proceedings of the 29th ACM Conference on User Modeling, Adaptation and Personalization (pp. 262-266).

---

### Official Review · Reviewer_Gn3Q · 2023-10-30

**Soundness:** 2 fair
**Presentation:** 3 good
**Contribution:** 2 fair
**Rating:** 3
**Confidence:** 4

**Summary:**

The paper primarily explores the integration of offline-RL with attention-based RS to enhance performance by incorporating user feedback as rewards. Unlike previous studies, the authors address challenges like off-policy training and data imbalances. They merge CQL and CL strategies to improve the current attention-based sequential recommendation and achieve better performance.

**Strengths:**

1.The paper is well-structured and technically sound, making it easy to follow.
2.The experiments conducted within the paper are detailed and thorough.
3.The proposed model improves sequential recommendation performance compared to several SOTA baselines, indicating the proposed modules can capture user interest better.
4.The authors try to address important topics, like off-policy training and data imbalances, in the field of offline-RL recommendation systems.

**Weaknesses:**

1. Some works related to contrastive learning in SR aren't mentioned, such as [1][2][3][4].
2. The main idea seems to merge two known methods, CQL (2020) and CL in RS (2020). This might not be fresh enough for ICLR standards.
3.Section 3.4 mentions that contrastive learning is frequently employed in past research. However, it's unclear how the method here differs from others, such as CL4Rec. Additionally, the term "temporal augmentation" in section title is introduced without adequate explanation.
4.Despite the paper presenting extensive experimental results, it lacks sufficient explanation and analysis of these findings, like Q3 in Section 4.

**Questions:**

1. The results show that SARSRec-CO and SASRec-CCQL perform similarly on the RetailRocket dataset. Does this mean CQL didn't make a big difference on this dataset? However, on the RC15 dataset, CQL seems more effective. Can the authors explain this?
2. Why did SASRec-CCQL not perform as well on the HR@10 metric for the RC15 Click dataset? Can the authors give more insight on this?
3. CL4Rec doesn't do as well as SASR-Rec on the Movielens and Amazon Food datasets. This is different from findings in original paper titled "Contrastive Learning for Sequential Recommendation [5]". Can the authors clarify why?
4.In Table 3, SARSRec-CO achieves the top HR@20 results. However, these aren't emphasized in bold.

[1] Chen, Yongjun, et al. "Intent contrastive learning for sequential recommendation." Proceedings of the ACM Web Conference 2022. 2022.
[2] Qiu, Ruihong, et al. "Contrastive learning for representation degeneration problem in sequential recommendation." Proceedings of the fifteenth ACM international conference on web search and data mining. 2022.
[3] Wang, Chenyang, et al. "Sequential recommendation with multiple contrast signals." ACM Transactions on Information Systems41.1 (2023): 1-27.
[4] Huang, Chengkai, et al. "Dual Contrastive Transformer for Hierarchical Preference Modeling in Sequential Recommendation." Proceedings of the 46th International ACM SIGIR Conference on Research and Development in Information Retrieval. 2023.
[5] Xie, Xu, et al. "Contrastive learning for sequential recommendation." 2022 IEEE 38th international conference on data engineering (ICDE). IEEE, 2022.

---

### Official Review · Reviewer_jovo · 2023-10-30

**Soundness:** 3 good
**Presentation:** 2 fair
**Contribution:** 3 good
**Rating:** 5
**Confidence:** 3

**Summary:**

The author's main objective is to use reinforcement learning to personalize sequential recommendations. To achieve this goal they propose several improvements to previous techniques. These improvements include: function approximation optimization using a contrastive learning objective, more conservative negative action sampling corrections, and highlighting the importance of progressive validation when analyzing model performance. To demonstrate the empirical impact of their contributions the authors use 5 datasets from the literature and baseline with four recently published algorithms in the literature. The result of their experiments show that their modifications give performance improvements across the board.

**Strengths:**

* The proposed work considers an important area of work with the advent of sequence models
* The authors consider a wide range of datasets from the literature
* The authors compare against several relevant algorithmic baselines and achieve significant improvements

**Weaknesses:**

* The paper is a little hard to follow throughout. This is true of both the prose and notation. One clear example is the $y$ variable which appears to have multiple meanings and indexes. Another example is the paper title which makes no mention of sequential recommendations even though this is the primary reason for the paper.
* The explanation that the SNQN and SA2C perform poorly due to divergence is not clear to me. For example, how was learning rate handled in these methods. Could the drop in performance not be explained by a learning rate that was not decayed quickly enough?
*  There does not appear to be a clear argument for the necessity of progressive validation despite this being listed as one of the main contributions in the introduction.

**Questions:**

* How was learning rate handled for the SNQN and SA2C baselines?
* Why do the different methods have different numbers of iterations in the experiment plots?
* Why are the curves almost identical for the various methods in the experiment plots? I would expect strong correlation but not identical.

**Details Of Ethics Concerns:**

I have no ethical concerns.

---

### Author Response · Authors · 2023-11-22
**Response to reviewers and decision to withdraw**

Dear reviewers,

We thank you for your insightful comments. I order to best address the points you have raised in your reviews, we have decided to withdraw the submission. That will allow us more time to improve it.